# Poxviruses package viral redox proteins in lateral bodies and modulate the host oxidative response

**Susanna R. Bidgood[1], Jerzy Samolej[2], Karel Novy[3], Abigail Collopy[1], David Albrecht[1], Melanie Krause[1], Jemima J. Burden[1], Bernd Wollscheid[3,4]\*, Jason Mercer[1,2]\***

**1** MRC Laboratory for Molecular Cell Biology, University College London, London, United Kingdom, **2** Institute of Microbiology and Infection, School of Biosciences, University of Birmingham, Birmingham, United Kingdom, **3** Swiss Federal Institute of Technology (ETH Zürich), Department of Health Sciences and Technology (D-HEST), Institute of Translational Medicine (ITM), Zürich, Switzerland, **4** Swiss Institute of Bioinformatics (SIB), Lausanne, Switzerland

\* bernd.wollscheid@hest.ethz.ch(BW); j.p.mercer@bham.ac.uk (JM)

## Abstract

All poxviruses contain a set of proteinaceous structures termed lateral bodies (LB) that deliver viral effector proteins into the host cytosol during virus entry. To date, the spatial proteotype of LBs remains unknown. Using the prototypic poxvirus, vaccinia virus (VACV), we employed a quantitative comparative mass spectrometry strategy to determine the poxvirus LB proteome. We identified a large population of candidate cellular proteins, the majority being mitochondrial, and 15 candidate viral LB proteins. Strikingly, one-third of these are VACV redox proteins whose LB residency could be confirmed using super-resolution microscopy. We show that VACV infection exerts an anti-oxidative effect on host cells and that artificial induction of oxidative stress impacts early and late gene expression as well as virion production. Using targeted repression and/or deletion viruses we found that deletion of individual LB-redox proteins was insufficient for host redox modulation suggesting there may be functional redundancy. In addition to defining the spatial proteotype of VACV LBs, these findings implicate poxvirus redox proteins as potential modulators of host oxidative anti-viral responses and provide a solid starting point for future investigations into the role of LB resident proteins in host immunomodulation.

## Author summary

Vaccinia virus is the prototype member of the Poxviridae family, whose members include monkeypox and variola virus, the causative agent of smallpox. All poxvirus particles, including vaccinia, are large and complex containing upwards of 70 different viral proteins.

Virions are composed of 3 main viral sub-structures: cores which house the viral genome, two proteinaceous lateral bodies thought to deliver host modulatory proteins, and the viral membrane that harbors proteins required for virus structure, binding and entry. While the protein composition of VACV cores and membranes is well known, for

**Data Availability Statement:** The datasets generated during and/or analysed during this study are available in the MassIVE repository under MassIVE ID MSV000086675_reviewer. Data can

**Funding:** This work was funded by a Sir Henry Wellcome Post-doctoral Fellowship (WT106080/Z/14/Z; to SRB), MRC Programme grant (MC_UU_00012/7; to JM), the MRC (MR/K015826/1; to JM), the European Research Council (649101, UbiProPox; to JM), and the Swiss National Science Foundation (grant 31003A_160259; to B.W.) and a Marie Skłodowska-Curie fellowship funded by the European Union (750673; to DA). JJB is supported by core funding to the LMCB (MC_U12266B). SRB recieved salary from the Wellcome Trust and DA from the European Union. The funders had no role in study design, data collection and analysis, decision to publish, or preparation of the manuscript.

**Competing interests:** The authors have declared that no competing interests exist.

technical reasons the collection of viral and cell proteins that reside in lateral bodies remains undefined. Here, we used a combination of controlled particle degradation and quantitative comparative mass spectrometry to identify viral and cellular candidate lateral body proteins. This revealed a large compendium of candidate lateral body cell and viral proteins for future analysis and uncovered viral redox proteins as a potential new class of poxvirus immunomodulators that are delivered to host cells via lateral bodies.

## Introduction

The main goal of all viruses is to transfer a replication competent genome from one host cell to another [1,2]. This involves the delivery of the viral genome and accessory proteins to the cytosol or nucleus. In doing so they trigger cellular innate immune responses, a vast network of receptors and signalling cascades designed to prevent pathogen infection and replication [3,4]. In turn viruses have evolved numerous mechanisms to evade and manipulate these host immune defences [5,6]. While most viruses appear to do this through the expression of nascent proteins during replication, both alphaherpesviruses and poxviruses have been reported to package and subsequently deliver accessory proteins with immunomodulatory capacity [7–11].

Herpesviruses do this using tegument, a protein layer between the capsid and membrane, which slowly dissociates during transit of viral capsids through the cytoplasm *en route* to the nucleus [9,12–15]. Poxviruses achieve this through two proteinaceous viral substructures termed lateral bodies [16–18]. Structurally, pox virions are brick-shaped particles composed of three main substructures: the viral core which houses the dsDNA genome, the lateral bodies (LBs) which reside on either side of the core, and the viral membrane that encapsulates these structures [10,19].

Vaccinia virus (VACV), the prototypic poxvirus, can enter cells by macropinocytosis or direct fusion at the plasma membrane [20–25]. In both cases fusion of viral and cell membranes results in deposition of viral cores into the cytoplasm where they undergo primary uncoating characterized by core expansion and the initiation of early gene transcription [16,26]. The LBs, which detach from the cores upon fusion, are deposited into the cytoplasm where they take on an effector function [27].

To date, attempts to define the complete composition of LBs and the function of their various protein constituents have been confounded by the molecular complexity of poxviruses (composed of more than 80 different viral proteins), and the inability to isolate intact LBs from viral particles or infected cells [8,18,28]. Recently, we identified the first 3 *bona fide* poxvirus LB proteins: An abundant viral phosphoprotein F17R, the viral phosphatase H1, and a viral glutaredoxin-2, G4L [8]. During late infection F17R acts to dysregulate (mTOR) and suppress interferon-stimulated gene responses [29,30]. As partial degradation of F17R mediates the release of the viral phosphatase H1L [8], we presume that F17R serves as the LB structural protein although any early host-modulatory role has yet to be investigated. The H1L phosphatase inhibits interferon-$\gamma$ driven, anti-viral immunity through dephosphorylation of the transcription factor STAT1 [31,32]. This activity of H1L is dependent on proteasome-mediated disassembly of LBs and independent of viral early gene expression [8]. G4L is a functional glutaredoxin whose activity is required for the formation of viral disulphide bonds during virion assembly [33–35]. G4L is expressed late, packaged into LBs but has not been assigned any host-modulatory activity [8,34].

Nevertheless, the generation of cellular reactive oxygen and nitrogen species (RONS) is an important component of cellular defense against incoming pathogens [36]. All viral infections cause a redox imbalance in the host cell, the outcome of which depends on the duration and magnitude of this imbalance [37,38]. Most commonly a pro-oxidative effect is triggered, which initiates anti-viral responses including autophagy, cell death, mTOR inhibition and RONS-coupled pattern recognition receptor signalling [39]. While some viruses (vesicular stomatis virus) [40,41] are restricted by this, several have evolved ways to harness the oxidative environment (influenza, respiratory syncytial virus, and human immunodeficiency virus) [42–48], and others to combat it (human cytomegalovirus) [49,50].

As a family of viruses that replicate exclusively in the cytoplasm, poxvirus replication is not surprisingly sensitive to RONS. IFNγ-driven nitric oxide (NO) production was shown to inhibit VACV and ectromelia virus replication in mouse macrophages [51], and NOS2-deficient mice are highly susceptible to ectromelia despite no known loss of ectromelia-targeted immune responses [52]. These reports suggest that poxviruses must overcome RONS for successful replication. Interestingly, VACV encodes a set of redox proteins including a human homolog of glutaredoxin-1 (O2L), the previously mentioned glutaredoxin G4L which is a homolog of human glutaredoxin-2, a SOD-like dismutase (A45R), E10R, a CxxC motif containing member of the ERV1/ALR family of eukaryotic redox proteins [53], a thiol oxidoreductase (A2.5L), and a protein with 2 putative thioreductase CXXC motifs (A19L) [34,54–57]. While no immunomodulatory role has been elucidated, VACV mutants deleted for these various proteins indicate that E10R, A2.5L and G4L are required for cytoplasmic disulphide bond formation and virion assembly [33,35,56,58], and A19L for virus transcription and assembly [57,59]. Deletion of either A45R or O2L suggests that they are dispensable for VACV replication [55,60].

Using controlled degradation of virions in combination with quantitative mass spectrometry (MS)-based proteotyping strategies we define the viral and cellular candidate proteins that constitute the VACV LB proteome. In addition to numerous cellular proteins, 15 candidate viral LB proteins, including 5 of 6 VACV-encoded redox proteins were identified. Follow-up analysis of the impact of ROS on VACV infection indicated that VACV exerts an anti-oxidative effect on host cells and that high ROS levels block viral replication. We further show that deletion or suppression of individual LB-resident redox proteins does not alter VACVs ability to suppress host redox responses to infection, suggesting that multiple LB-redox proteins may be required. In addition to elucidating the protein composition of poxvirus LBs, these findings further suggest that VACV LBs act as early immunomodulatory delivery packets and provide a solid starting point for additional studies regarding the role of poxvirus redox proteins in modulating host oxidative responses to infection.

## Results

### Defining the poxvirus lateral body spatial proteotype

With a few exceptions, the compendium of proteins that make up poxvirus LBs remain unknown [7,8]. To define and investigate the LB proteome we employed sub-viral fractionation of VACV mature virions (MVs) in combination with quantitative MS-based analysis (illustrated in Fig 1A). Using modified biochemical protocols first established by Ichihashi *et al.* [17], purified intact MVs were stripped of their membranes using detergent and reducing agents (NP40 and DTT). The viral LB-core fraction was isolated away from the soluble membrane fraction by centrifugation. After additional wash steps, the LB-core fractions were then subjected to controlled proteolysis, using varying concentrations of trypsin, to remove/solubilize LB components while minimizing core protein degradation. The LB-free cores were then

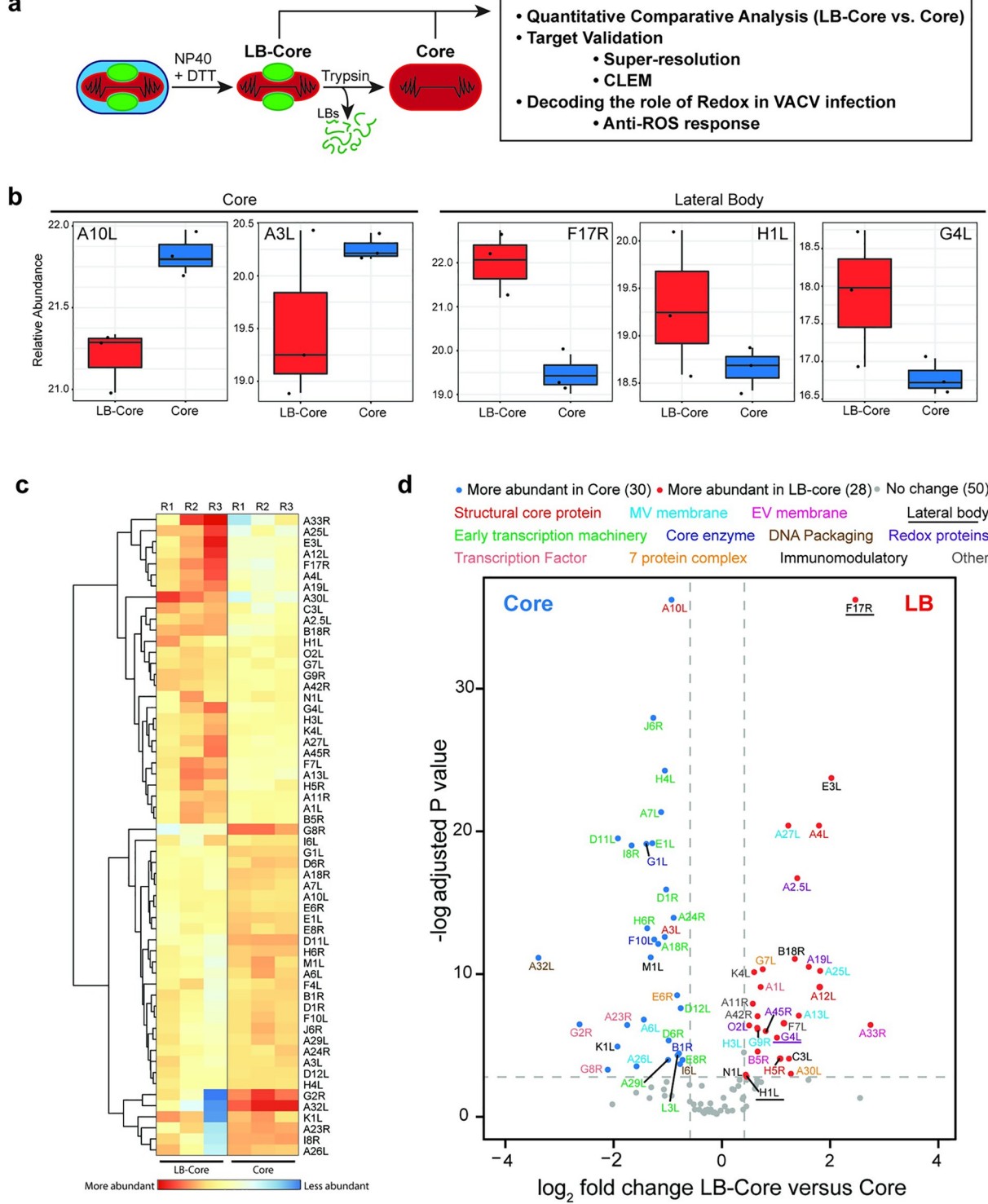

**Fig 1. Defining the VACV LB spatial proteotype. (a)** Schematic of controlled degradation of purified VACV MVs used in this study (details in Materials and Methods). **(b)** Relative abundance plots of known core (A10L, A3L) and LB (F17R, H1L, G4L) proteins across LB-Core and Core fractions. **(c)** Heat map showing the relative abundance of VACV proteins between LB-Core and Core fractions. **(d)** Relative abundance of viral proteins in LB-Core versus Core fractions. Proteins are colour coded with regard to LB-Core vs. Core abundance: more (red), less (blue) or no difference (grey). Experiments were performed in triplicate and significance scored as a minimum of two-fold greater in LB-Core vs. Core samples. An adjusted p-value ≤ 0.08 cut off was set using the adjusted p-value of the *bone fide* VACV LB protein H1L (c-d).

washed and isolated by centrifugation. When samples were subjected to immunoblot analysis for membrane (A17L), LB (F17R) and inner core (A10L) proteins, 125ng/ml trypsin appeared to provide optimal LB removal while maintaining core integrity (S1A Fig). When carried over to large-scale preparations for comparative quantitative MS this fractionation protocol resulted in 97% membrane removal, and 98% LB removal with only a 3% loss in core integrity (S1B Fig). Electron microscopy (EM) confirmed that the purified virions (whole VACV) that we started with were free from contaminating cell debris or organelles, and that the protocol provides two enriched sub-virion fractions: membrane-free cores with associated LBs (Pellet 1; LB-Core), and intact cores in which the LBs have been effectively removed (Pellet 2; Core) (S1C Fig).

For spatial proteotype analysis, enriched LB-Core and Core samples were subjected to lysC-assisted tryptic digestion under pressure-cycling conditions [61,62]. Samples were analysed by data-dependent acquisition liquid chromatography mass spectrometry (LC-MS/MS) and quantified across the two conditions. Consistent with previous reports, a total of 108 viral proteins were identified amongst which 49 of 50 VACV core components were identified [63] (Sheet 1 in S1 Table). For quality control, the abundance of *bona fide* core (A10L, A3L) and LB (F17R, H1L and G4L) components were compared across samples (Fig 1B). The relative abundance of A10L and A3L was enriched in core samples while F17R, H1L and G4L were enriched in LB-core samples. Using the low abundance LB protein H1L as a cut off [8,64], 50 proteins, showed no change in abundance between the two samples while 58 proteins were significantly enriched in either LB-Core or Core samples (Sheet 2 in S1 Table). Hierarchical clustering confirmed that these 58 were consistently enriched in LB-Core or Core samples across individual replicates (Fig 1C). Amongst these, 30 proteins were associated with VACV cores and 28 potential LB candidates emerged (Fig 1C and 1D and Sheets 3 and 4 in S1 and S2 Tables).

Five subsets of expected core enriched proteins were identified: 2 structural core proteins (red), 15 viral early transcription machinery proteins (green), 2 DNA packaging proteins (brown), 3 core enzymes (navy), and 3 viral intermediate and late transcription factors (coral) (Fig 1C and Sheet 4 in S1 Table). Five of the identified proteins have not been previously associated with VACV cores: VACV seven protein complex protein E6R (orange), VACV membrane proteins A26L and A6L (cyan), and the immunomodulatory proteins M1L and K1L (black). The remaining 19 known core proteins (grey) showed no significant difference in abundance between samples, suggesting that LB removal did impact their core association (Fig 1D).

The proteins enriched in LBs included the previously described LB proteins F17R, H1L and G4L (underlined), 6 immunomodulatory proteins (black), 5 putative redox modulating proteins (purple) and 2 seven protein complex proteins (orange) (Fig 1D and Sheet 3 in S1 and S3 Tables). We also detected 3 core proteins (red), 5 MV membrane proteins (cyan), 2 EV membrane proteins (pink), 1 late transcription factor A1L (coral), and 4 proteins that did not fall into a definable category (grey): 1 protein required for crescent formation (A11R), the VACV nicking/joining enzyme K4L, VACV profilin A42R and 1 protein with no defined function F7L (S2 Table).

Visualizing our findings in light of the VACV MV protein-protein interaction network uncovered by Mirzakhanyan and Gershon [65], we built a schematic of our candidate LB proteins and their proposed interactions (S2 Fig and S2 Table). The extensive connections between the identified proteins likely explains the presence of 'contaminating' VACV membrane proteins A6 and A26 in the core proteome, and membrane and core components within the LB proteome. Taken together, as the sub-viral localization of 13 of these proteins has been defined as MV/EV membrane, core and LB [8,10], we were left with 15 novel LB candidate

proteins (A1L, A2.5L, A11R, A19L, A30L, A42R, A45R, B18R, C3L, E3L, F7L, G7L, K4L, N1L, O2L).

## Host proteins reside in VACV LBs

As the VACV MVs used for these studies were purified from human cells, we had the opportunity to extend our analysis of the VACV LB proteome to include human proteins. Doing so, we identified a total of 586 human proteins associated with VACV virions (S1 and S2 Tables). While we confirmed 78% [66], 67% [67] and 80% [68] of the VACV-associated cellular proteins found in previous MS studies, improved MS methodology and detection allowed us to identify many more. Of the 586 cellular proteins detected, 210 were significantly enriched in LB-Core or Core samples (S2 and S3 Tables), while the remaining 376 proteins were not enriched in either. Hierarchical clustering across individual replicates showed that amongst the 210 enriched proteins, 93 were consistently found in LB-Core samples and 117 in Core samples (Fig 2A and 2B and Sheets 3 and 4 in S3 Table). In the comparative quantitative analysis, 6 human proteins (mitochondrial transcription factor A, vimentin, Ankyrin repeat domain-containing protein 17, NADH dehydrogenase 1 alpha subcomplex subunit 7, X chromosome RNA-binding motif protein, mitochondrial 39S ribosomal protein L14) were more significantly enriched at a greater abundance than the major LB viral protein F17R, suggesting that these human proteins may be major LB constituents.

String visualization of the human candidate LB proteins revealed 7 major protein interaction clusters (Fig 2C and Sheet 3 in S3 Table) [69]. These included 9 ribosome subunits, 6 proteins involved in splicing and 16 cytoskeletal-associated proteins including regulators of actin dynamics, myosins and intermediate filaments (Fig 2 and Sheet 3 in S3 Table). Strikingly, over half [45] of the candidate proteins were of mitochondrial origin including 15 proteins of the respiratory chain, 15 mitochondrial ribosomal subunits, 5 mitochondrial enzymes and 6 proteins involved in mitochondrial membrane stability and import.

While the purity of our virus preparations was verified by EM (S1C Fig), to assure that the identified cell factors represent genuine LB candidates and not just contaminating cellular material, we performed control virus purifications from uninfected and infected cell lysates. Immunoblot analysis for cellular proteins not detected in LB fractions (actin and tubulin) and those detected in LB fractions [histone 1, 60S ribosomal protein L17 (RPL17), superoxide dismutase 1 (SOD1) and translocase of outer mitochondrial membrane 20 (Tomm20) was performed after each purification step: cell lysis (nuclear pellet/supernatant), pellet from 36% sucrose cushion, virus band from 25–40% sucrose gradient, and final virus pellet (S3 Fig).

In uninfected cell purifications, actin, tubulin, histone 1, SOD1 and Tomm20 were no longer detectable after the 36% sucrose cushion, while RPL17 was carried through to the final pellet (S3 Fig; uninfected). In infected cell purifications, virus (A10) was concentrated enough for detection in the 36% sucrose cushion pellet. As expected, tubulin and actin were not detected in the final virus pellet. Apart from histone 1, candidate LB proteins RPL17 (to reduced amounts), SOD1 and Tomm20 were found in detectable amounts in the final virus pellet (S3 Fig; infected). Taken together with the EM, these results suggest that the cellular candidate LB proteins are not co-purification contaminants but associated with purified virus particles.

To define the localization of these cell proteins within virions, purified VACV was subjected to fractionation and LB degradation (outlined in Fig 1A). Fractionation was confirmed by immunoblotting for core (A10), membrane (A17) and LB (F17) components (Fig 3A). Upon fractionation most of the membrane (A17) was separated from the LB-Core complex containing A10 and F17. Upon trypsinization of the LB-Core complex, the LB (F17) was lost while the cores (A10) could be recovered by centrifugation (Fig 3A).

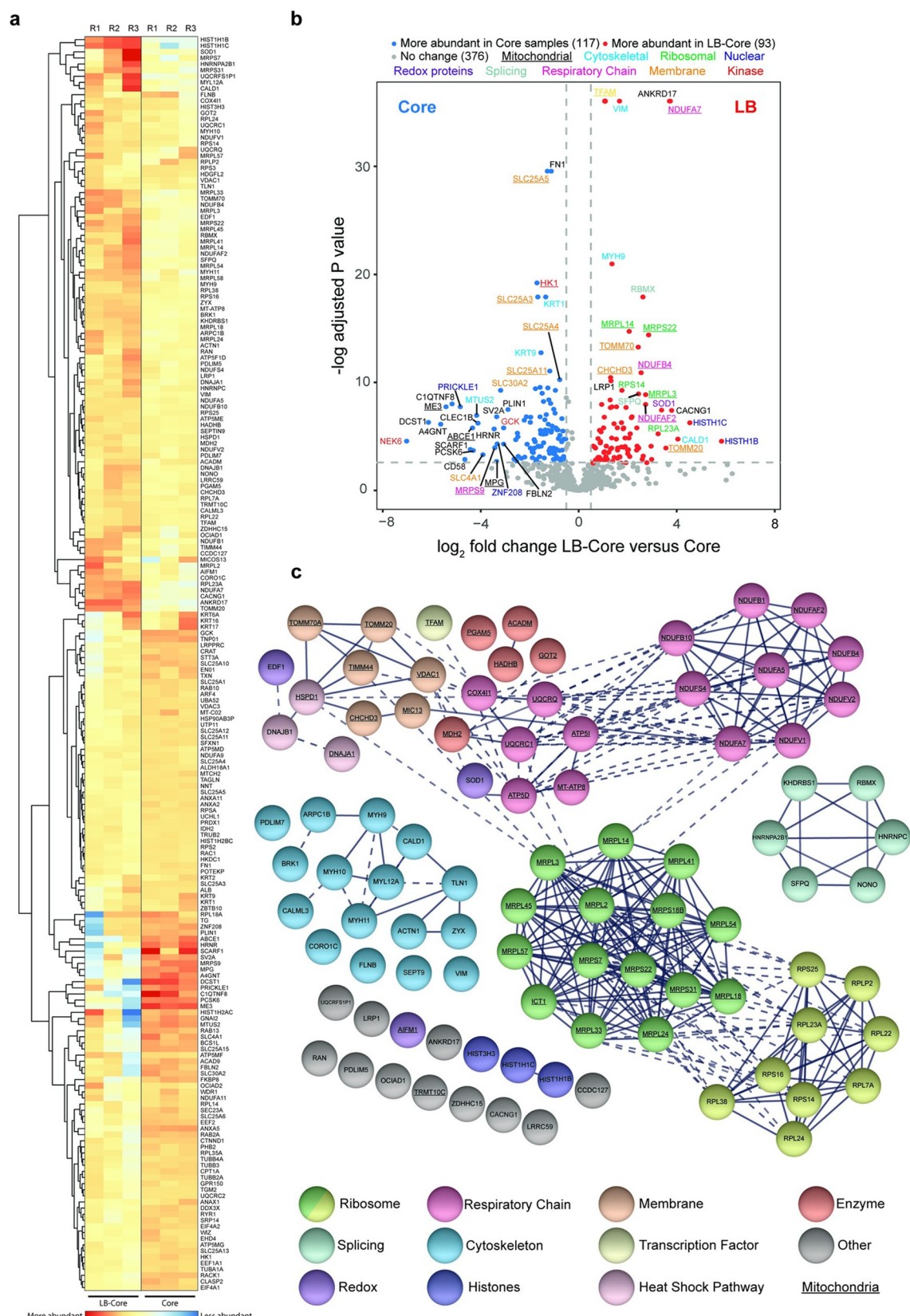

**Fig 2. Candidate cellular protein constituents of VACV lateral bodies.** (a) Heat map showing the relative abundance of human proteins in LB-Core vs. Core fractions. (b) Relative abundance of human proteins in LB-Core vs. Core fractions. Proteins are colour coded with regard to LB-Core vs. Core abundance: more (red), less (blue) or no difference (grey). Experiments were performed in triplicate and significance scored as a minimum of two-fold for LB-Core relative to Core with an adjusted p value ≤ 0.08 (a & b). (c) STRING protein-protein interaction network of candidate human proteins enriched in LB-Core vs. Core fractions. Proteins were clustered and coloured according to function as indicated. Proteins involved in mitochondrial function are underlined. Solid lines depict confirmed physical interaction (confidence >0.9), dashed line indicates suggested co-location in screening studies.

Regarding the cell proteins, upon fractionation Histone 1 and Tomm20 remained exclusively with the LB-Core fraction, while RPL17 and SOD1 fractionated into both LB-Core and membrane (Fig 3A). Upon trypsin treatment of the LB-Core fraction, Histone 1 and Tomm20 were completely degraded while a portion of RPL17 and SOD1 remained with the core pellet. To extend these findings, LB-Core complexes of fractionated WR EGFP-A4L virions were subjected to immunofluorescence staining for RPL17 or Tomm20 and imaged using dual-colour SIM (Fig 3B). In line with our EM imaging (S1C Fig), VACV cores (green) appeared expanded upon fractionation. Both RPL17 and Tomm20 (red) were found in LBs and could be visualized in sagittal (two LBs) and frontal (single LB) virion orientations (Fig 3B). Although we only tested a small subset of proteins, consistent with the quantitative MS and biochemical fractionation data, the cell proteins tested were enriched in LBs relative to cores. In addition, we show that Tomm20 and Hist1 biochemically fractionate akin to the *bona fide* LB protein F17, and that RPL17 and Tomm20 reside within LBs.

## VACV LBs harbour a set of viral proteins with redox modulating potential

The presence of so many mitochondrial proteins in LBs along with the co-identification of five putative viral redox proteins (G4L, A2.5L, A19L, A45R and O2L) prompted us to focus on these LB candidates. Briefly, with the exception of A45L, which is 39% identical to human Cu-Zn SOD [55], each of these proteins contain Cxx(x)C motifs characteristic of redox modulating enzymes [70]. Amongst these, the glutaredoxin G4L, a homologue of human Glutaredoxin-2, is the only previously identified LB protein [8]. Together with A2.5L and E10, which was previously shown to not reside in LBs [8], G4L is reported to be part of a VACV encoded cytoplasmic disulphide formation pathway that is essential for virus assembly [33–35,56]. The VACV protein A19L carries two CxxC motifs that are important for interaction with VACV core transcription machinery and are essential for VACV infection [57]. A45R is a non-functional SOD-like dismutase, whose *Leporipoxvirus* homologue has been shown to disrupt host SOD1 activity [55,71,72], and finally, the O2L protein is a homologue of human Glutaredoxin-1 that contains a single CxxC motif and displays *in vitro* thioltransferase activity [54,60,73].

Relative abundance plots of the MS data indicated that G4L, A2.5L, A19L, A45R and O2L were all significantly enriched in LB-core over core samples (Fig 3C). While indicative of LB residence, to validate the MS results and confirm the localization of these proteins to LBs we turned to super-resolution microscopy. We have previously used structured illumination microscopy (SIM), stimulated emission depletion (STED), and stochastic optical reconstruction microscopy (STORM), in combination with single particle averaging to map viral proteins to distinct VACV substructures including LBs, cores and the viral membrane [74–78].

To this end, we built a library of recombinant VACVs using the strategy employed for the successful identification of the major LB protein F17R [8]. A parental virus harbouring a mCherry-tagged version of the core protein A4L was used to generate a set of recombinant viruses that carry an additional EGFP-tagged version of G4L, A2.5L, A19L, A45R or O2L. Having shown that G4L is a LB protein using immuno-EM [8], we used it as a test case. Virions were imaged by dual-colour SIM and localization models of G4L and A4L were generated as

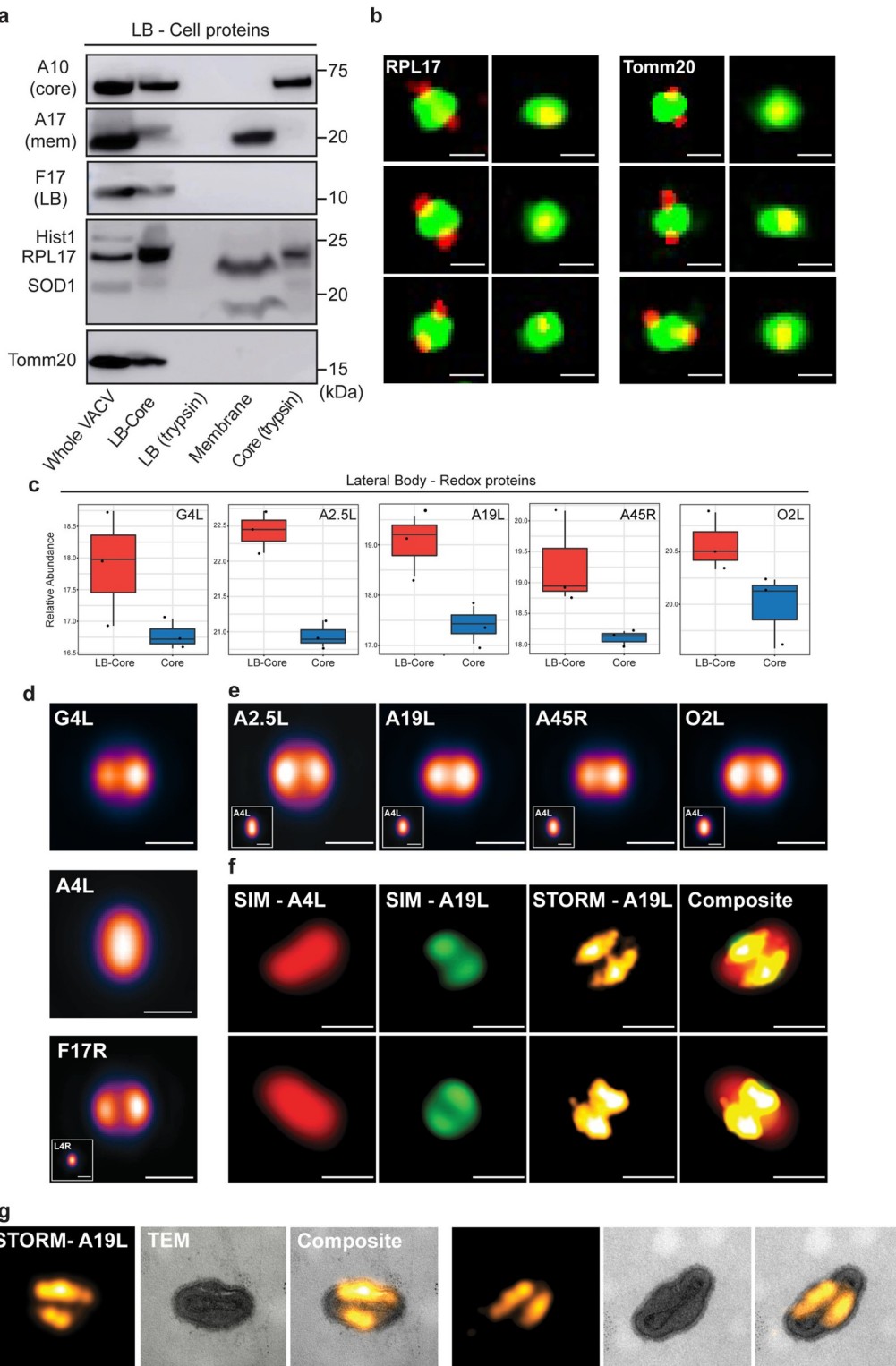

**Fig 3. Cell and viral redox proteins reside in LBs. (a)** Biochemical localization of candidate cellular LB proteins. MVs were subjected to controlled degradation as outlined in Fig 1A, and fractions subjected to immunoblot analysis for indicated viral and cellular proteins. Experiment was performed in triplicate and representative blots displayed. **(b)** SIM images (sagittal and frontal orientation) of RPL17 and Tomm20 localization in fractionated WR EGFP-A4L virions. Experiments were performed in duplicate and six representative virions displayed. **(c)** Relative abundance plots of the five

putative redox modulating viral enzymes (G4L, A2.5L, A19L, A45R, O2L) across LB-Core and Core fractions **(d)** VirusMapper models (sagittal orientation) of G4L (LB) and A4L (core) protein localization using WR mCherry-A4L G4L-EGFP virions (n = 949). A VirusMapper model of WR L4R-mCherry EGFP-F17R is displayed for comparison (n = 231) **(e)** VirusMapper models (sagittal orientation) of EGFP-tagged redox proteins: A2.5L, A19L, A45R, O2L (n > 100). **(f)** Correlative SIM-STORM imaging of WR mCherry-A4L A19L-EGFP virions. Two representative virions are displayed (Overview in S4A Fig) **(g)** Correlative STORM-TEM of WR mCherry-A4L A19L-EGFP virions immunolabelled with anti-GFP nanobody directly conjugated to AlexaFluor647-NHS. STORM images of A19L were registered with EM micrographs. Two representative virions are displayed (Overview S4B Fig). **(b, d-f)** Scale bars = 200 nm.

previously described [74,77]. Using A4L to identify virion position and orientation (Fig 3D; middle panel), the localization models showed that G4 resides in two elliptical structures at the sides of cores corresponding to VACV LBs (Fig 3D; top panel). A model of virions containing EGFP-tagged F17R, the major *bone fide* LB protein, is displayed for comparison (Fig 3D; bottom panel). Applying this imaging and analysis pipeline to the remaining VACV redox proteins we confirmed, consistent with the MS results, that A2.5L, A19L, A45R and O2L are each novel VACV LB proteins (Fig 3E).

To further exemplify the utility of this super-resolution virion protein mapping approach, we performed correlative SIM-STORM and correlative STORM-TEM (transmission electron microscopy) on mCherry-A4 A19L-EGFP recombinant VACV. For both, virions were permeabilised and immuno-labelled with AlexaFluor647-NHS conjugated anti-GFP nanobodies to enable STORM imaging of the A19L-EGFP. For SIM-STORM, virions were imaged using dual-colour SIM for core (A4L) and LB (A19L), followed by far-red STORM reporting on the fluorescently-labelled nanobodies. Representative virions displayed in Fig 3E (overview S4A Fig), show VACV cores (SIM-A4L) flanked by two A19L positive LBs (SIM-A19L). As anticipated, the A19L STORM reconstructions largely overlap with the SIM A19L images, albeit with better resolution (Fig 3F; STORM-A19L). For STORM-TEM, virions were first imaged by STORM for A19L-EGFP, followed by EM sample preparation, sectioning and imaging of the same virions by TEM. Correlation of these images supports the SIM and STORM data, confirming that A19L resides in VACV LBs (Fig 3G; overview S4 Fig).

Together these results confirm our MS data, which identifies viral redox proteins as a new class of VACV LB constituents and demonstrate the utility of correlative super-resolution imaging to define the sub-viral location of proteins in poxviruses.

## VACV infection exerts an anti-oxidative effect on host cells

All virus infections exert a redox imbalance in the host cell and intracellular RONS production. This in turn activates anti-viral immune responses such as the upregulation of autophagy, promotion of cell death, inhibition of mTOR and Pattern Recognition Receptor downstream signalling events [39]. Given the presence of five putative redox modulating enzymes in LBs we considered the possibility that VACV modulates cellular RONS production for successful infection. To assess this, we first looked at the impact of VACV infection on reactive oxygen species (ROS) levels generated during normal cellular respiration. Cells were infected with WT VACV and the percentage of ROS positive cells quantified at 1 h and 8 h post infection (pi) using an oxidant-sensitive probe in combination with flow cytometry. These time points correspond to early infection, when LB proteins are delivered into the cytoplasm by incoming virions, and late infection when greater amounts of these LB proteins are newly synthesized. Untreated, uninfected cells served as the negative control and uninfected cells treated with tert-butyl hydroperoxide (TBHP), a ROS inducer, were included as a positive control for ROS activation. At both time points ~40% of untreated, uninfected control cells were ROS positive (Fig 4A). Infection

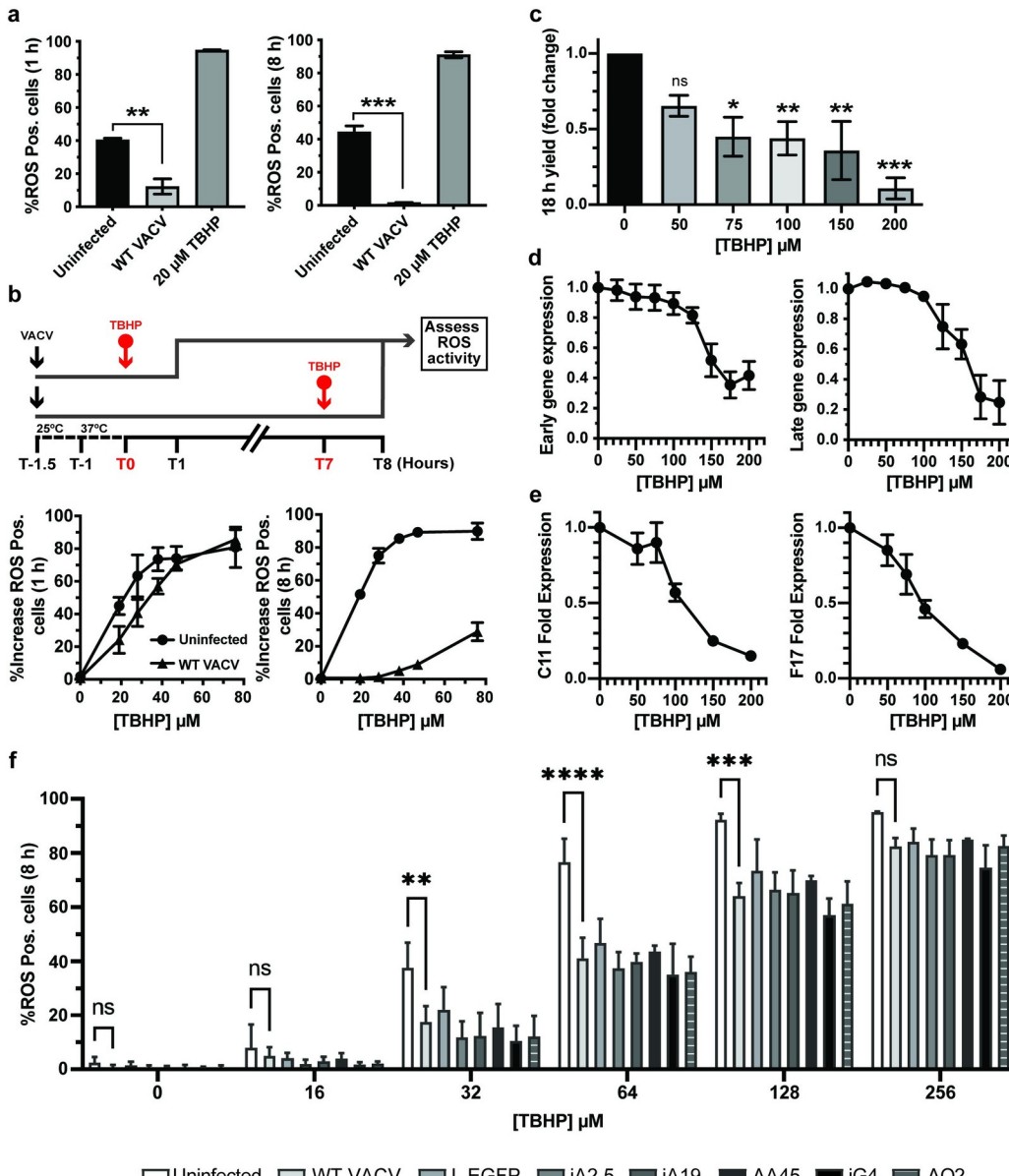

**Fig 4. VACV supresses host cell oxidative responses to promote infection. (a)** The percentage of ROS activated A549 cells measured at 1 and 8 h pi in cells left uninfected (black bar) or infected with WT VACV (Light grey bar). As a positive control, uninfected cells were treated with 20 µM TBHP for the final hour of the experiment (dark grey bars). **(b)** Top: schematic of experiment to assess impact of VACV infection on ROS activation. VACV was bound to A549 cells for 30 min at RT prior to being shifted to 37˚C for 1 h. Cells were challenged with the indicated concentrations of ROS inducer TBHP for 1 h prior to quantification of ROS-positive cells at 1 h pi and 8 h pi. Bottom: The percentage of uninfected (circles) and infected (triangles) ROS positive cells at 1 h pi and 8 h pi challenged for the final hour with the indicated concentrations of TBHP (n = 3; ± SEM). **(c)** A549 cells were infected with WT WR VACV (MOI = 1) in a range of TBHP concentrations. At 18 h pi, the infectious virus yield was assessed by plaque assay (n = 3; mean ± SEM). **(d)** A549 cells infected with WR E-EGFP or L-EGFP were challenged with the indicated concentrations of TBHP for 8 h. Early and late viral gene expression was assayed by flow cytometry and the data plotted as the fold change from infected untreated cells (n = 3; mean ± SEM). **(e)** A549 cells infected with WT VACV were challenged with the indicated concentrations of TBHP for 8 h. Total RNA was extracted and early (C11R) and late (F17R) gene expression assessed by RT-qPCR. Data is displayed as fold expression normalised to GAPDH (n = 3 ± SEM). **(f)** Following the schematic in (b), the percentage of ROS activated A549 cells was measured using MitoSOX Red at 8 hpi in uninfected (white bar) or WT, WR L-EGFP, iA2.5, iA19, ΔA45, iG4, or ΔO2 infected cells and treated with indicated concentrations of TBHP for the final hour of the experiment (n = 3 ± SD). (a, c, f) *P<0.05, **P<0.005, ***P<0.0005.

with VACV resulted in a >3-fold reduction in ROS positive cells at 1 h pi (12%), and a >23-fold reduction by 8 h pi (1.7%) (Fig 4A).

To test how effective VACV is at shunting ROS activation, cells infected with WT VACV were challenged with increasing concentrations of TBHP at 0 h or 7 h pi and the increase in ROS positive cells determined at 1 h and 8 h pi (Illustrated in Fig 4B; top). Compared to uninfected cells, at 1 h pi VACV infection reduced the number of ROS positive cells by 1.84-fold at low TBHP concentrations prior to being overcome at concentrations $\geq$ 50 μM (Fig 4B; lower left). At 8 h pi, VACV infection partially attenuated ROS activation by TBHP (10-fold at 50 μM relative to uninfected controls, with a 28.8% break through at 80 μM) (Fig 4B, lower right). These results indicated that VACV exerts a modest anti-oxidant effect on host cells both early and late during infection, a finding consistent with LB-mediated delivery of redox proteins and their subsequent post-replicative expression.

## High levels of ROS are detrimental to VACV infection

Despite the fact that VACV encodes and packages 5 proteins with redox modulating potential, we found that the artificial induction of high ROS levels could not be fully supressed during infection. This prompted us to ask if elevated cellular ROS levels would be deleterious to VACV replication. Cells were infected with WT VACV in the presence of various concentrations of TBHP (50 μM- 200 μM) and the infectious yield determined at 18 h pi. We observed a dose-dependent effect with a 1.5-fold reduction in virus yield at 50 μM TBHP, and a 3.7-fold reduction at 200 μM TBHP (Fig 4C). While the effects were modest, LDH cytotoxicity assays of cells treated with the same concentrations of TBHP indicated that this reduction was not due to cell death (S5A Fig). To determine at what stage of VACV infection ROS was acting we measured VACV early and late gene expression using recombinant viruses that express EGFP under the control of endogenous early (WR E-EGFP) or late (WR L-EGFP) viral promoters. Consistent with the viral yield experiments, high concentrations of TBHP (>100μM) resulted in decreasing numbers of cells expressing early and late genes as determined by flow cytometry at 8 h pi (Fig 4D). LDH assays confirmed that this decrease was not due to cell death (S5B Fig). To further define if the impact on viral gene expression was at the level of transcription or translation, cells were infected with WT VACV in the presence of increasing concentrations of TBHP. At 8 h pi RT-qPCR was used to determine the vRNA levels of the VACV early gene (C11L) and a VACV late gene (F17R) [79]. A moderate but consistent dose-dependent inhibition of both early and late viral gene transcription was seen with increasing TBHP concentrations (Fig 4E). This indicated that the reduction in VACV gene expression seen upon increasing oxidative stress occurs at the level of VACV transcription or viral transcript stability.

As discussed above, A2.5, A19 and G4 are each required for virus assembly while A45 and O2 are not required for VACV replication in tissue culture [33–35, 54–57,59,60]. Using inducible A2.5 (iA2.5L), A19 (iA19L), G4 (iG4L) and deletion A45 (ΔA45R) and O2 (ΔO2L) viruses, we asked if any individual LB redox protein was responsible for VACVs diminution of TBHP-induced ROS production (Fig 4B). As outlined in Fig 4B (top) cells were infected with WT VACV, iA2.5L, iA19L, iG4L - in the absence of inducer- ΔA45R or ΔO2L and challenged with various concentrations of TBHP for 1h prior to assessing ROS activity. This assay was restricted to the 8h timepoint as despite not making A2.5, A19 and G4 in the absence of inducer these proteins are packaged within the permissive iA2.5L, iA19L, iG4L virions used for infection. Uninfected, TBHP-treated cells served as controls for ROS activation and since A45R and O2L ORFs were replaced by EGFP, a recombinant virus that expresses EGFP from a late virus promoter (VACV L EGFP) was included to exclude any off-target effects of EGFP

expression on ROS activation/detection. Addition of TBHP to uninfected samples resulted in a dose-dependent increase in the percent of ROS-positive cells ranging from 2% in untreated cells to >90% at 128μM and 256μM TBHP (Fig 4F). Infection with WT VACV significantly reduced the percentage of ROS-positivity by 53%, 46% and 30% in cells treated with 32, 64 and 128μM TBHP, respectively (Fig 4F). Infections with iA2.5L, iA19L, or iG4L in the absence of inducer, ΔA45R or ΔO2L showed no impairment in their ability to supress TBHP-mediated ROS activation (Fig 4F). This indicates that VACVs ability to dampen the host oxidative response does not depend on a single LB redox protein. Collectively, these results show that VACV infection down regulates cellular ROS production, and that high levels of ROS impact VACV transcription or transcript stability, resulting in impaired or delayed protein expression and a moderate impact on virion production. That no single LB redox protein was found to be responsible for damping cellular ROS, suggests that there may be some level of functional redundancy between the 5 proteins.

## Discussion

All members of the *Poxviridae* carry LBs between their membrane and core [19]. Using EM, Dales first observed and described LBs as 'lateral dense elements' which disassociate from the virus core during entry [16]. *In vitro* experiments on purified VACV virions soon followed, showing that LBs remain associated with viral cores upon biochemical removal of the virion membrane [18]. Together these findings suggested the poxviruses were composed of multiple structural elements and hinted that LBs were independent entities. *In vitro* studies by Ichihashi and Oie confirmed that LBs were distinct from virus cores and composed of proteins [17].

Using similar techniques, we identified the first three *bona fide* viral LB proteins [8]: F17, a small basic VACV phosphoprotein, the VACV phosphatase H1, and G4, a viral glutaredoxin required for cytoplasmic disulphide formation during virion assembly [33]. We showed that F17 was the major LB protein and that it is degraded in a proteosome-dependent manner to release H1 during deposition of virus cores into the cytoplasm. Upon release H1 fulfils its effector function—blocking STAT1 nuclear translocation and preventing STAT1-mediated anti-viral responses—prior to early gene expression. Based on these findings we speculated that LBs are akin to Herpesvirus tegument, serving to deliver immune- or cell-modulating proteins into host cells, enabling the virus to establish it's replicative niche [8,80].

While LBs appear to play an essential role in the poxvirus lifecycle, the composition of LBs and the function of their constituents has never been fully defined. This is partially due to difficulties associated with isolation or purification of LBs from cells or virions, and that the majority of studies are from the pre-omics era. Taking advantage of spatial proteotyping, we could assess protein enrichment across biochemically fractionated virion samples (LB-Core vs. Core), thereby circumventing the need for LB isolation or purification [17,18].

This approach also provided us with the opportunity to ask, for the first time, if host proteins are packaged into VACV LBs. To this end, we identified 93 candidate human proteins enriched in LBs. Most proteins identified were cytoskeletal, ribosomal or mitochondrial in origin. Immunoblot analysis of purified intact and fractionated virions showed that a small subset of candidate cellular LB proteins (Hist1, RPL17 SOD1 and Tomm20) are not contaminants but components of purified virions (Fig 3A). We show that these proteins, with varying degrees, biochemically fractionate akin to the *bona fide* LB protein F17 [8]. Using SIM we further demonstrate that two of these proteins, RPL17 and Tomm20, both reside within LBs. That RPL17, and therefore presumably intact ribosomes, were found in the final pellet of uninfected "virion purifications", future assignment of ribosomal proteins as virions components should be verified by super-resolution imaging modalities. In addition to RPL17 and

Tomm20, we recently showed that vimentin—which localizes to viral replication sites and is packaged into virions [67,81]–also biochemically fractionates with Core-LBs, and like other LB candidates is trypsin sensitive [82]. Strikingly, 6 of the host proteins were found to be more abundant in LBs than F17 raising the possibility of active LB incorporation. Further confirmation and research into these host factors would be highly valuable and of future interest, as we speculate that the virus packages and delivers these factors to facilitate establishment of the VACV replicative niche.

While this is the first study to look at host proteins in VACV LBs, there are two previous reports using similar fractionation/degradation protocols to identify VACV structural components [17,83]. Though well performed, due to technical limitations at the time, one study lacks genomic information and the other is limited to N-terminal sequencing. This makes it difficult to compare the findings between one another or with our own. Nonetheless of the 6 viral proteins identified in the latter study; 3 described as core / core-LB are *bona fide* core proteins (A3, A10, A12), 2 described as membrane-LB are *bona fide* membrane proteins (A17, D8) and the remaining protein described as core-LB (G7) we identified in this study as a candidate LB component. As a member of the VACV 7-protein complex, G7 is a required for the association of viral membrane with the to-be intra-virion components during VACV assembly [84]. While it's tempting to speculate that the entire 7-protein complex may be within LBs [85,86], the only other member identified was A30, a known interactor of both F17 and G7 [87,88]. We did not identify other 7-protein complex members such as the F10 kinase, a finding that was consistent with its sub-viral compartmentalization during virion maturation [89]. It will be of future interest to determine if, in addition to their role in virion assembly, F17, G7 and A30 act as LB-delivered effectors in any capacity.

In this report we focused on confirming the LB localisation of five viral proteins with redox modulating capabilities; A2.5L, A19L, A45R, O2L and the previously identified G4 [8]. Super-resolution imaging was used to unequivocally show that these proteins are LB components. Together with the fact that VACV replicates exclusively in the cytoplasm, this strongly suggested that controlling the production of intracellular ROS is important for VACV replication. Consistent with LB delivery of redox modulators, VACV infection diminished basal cellular ROS levels as early as 1 h pi. We have previously shown that the reducing environment of the cytoplasm is essential for the dismantling of incoming disulphide-linked core proteins to allow for core expansion and subsequent early gene transcription [8]. Consistent with this, we show here that exogenous activation of ROS is detrimental to VACV infection prior to early gene transcription. Collectively this data suggests that, like cytomegalovirus, VACV has evolved to overcome the anti-viral oxidative challenge presented by cells during infection [49,50]. In support of this, IFN$\gamma$-driven NO production inhibits VACV and ectromelia virus replication in mouse macrophages [51], and NOS2-deficient mice are highly susceptible to ectromelia despite no known loss of ectromelia-targeted immune responses [52]. Taken together with our findings, these results suggest that poxviruses must overcome RONS for successful replication.

Previous work indicates that A45 and O2 are not required for VACV replication in tissue culture, that A45 is dispensable *in vivo* and that A2.5, G4 and A19 are each required for virus assembly [33–35,54–57,59,60]. While these attributes make investigating the individual factors challenging, we used a set of inducible / deletion viruses to investigate the impact of losing single LB redox proteins on VACVs ability to supress TBHP-activated ROS. That loss of individual proteins did not show an effect was not too surprising given the potential for functional redundancy garnered by packaging 5 redox proteins in LBs. Akin to the VACV strategy of dedicating 9 different yet redundant viral factors to inhibit the NF-kB pathway [90,91], we postulate that the various redox proteins target RONS signalling through divergent immunomodulatory targets.

In summary, by combining comparative quantitative MS, super-resolution microscopy and molecular biology we uncovered the candidate poxvirus LB proteome. Furthermore, we found a correlation between the packaging of viral redox proteins in LBs, the ability of VACV to disrupt the intracellular oxidative environment and the inhibition of VACV infection by ROS. The presence of 5 out of 6 VACV-encoded redox proteins in LBs underscores the potential importance of blocking RONS for viral replication. In addition to defining the compendium of cell and viral candidate proteins found in LBs as a valuable resource to the fields of virology and immunology, this study provides a solid foundation for mechanistic studies aimed at understanding the role of RONS in the cellular control of poxvirus infection and consequently how viruses have evolved to evade this.

## Materials and methods

### Cell lines and virus propagation

BSC40 (kind gift of P. Traktman, Medical University of South Carolina, Charleston, SC, USA), hTert RPE-1 (Clonetech laboratories), A549 (ATCC CCL-185) and HeLa cells (ATCC CCL-2) were grown in Dulbecco's modified Eagle's medium (DMEM) supplemented with 100 units/ml penicillin, 100 μg/ml streptomycin, 1 mM sodium pyruvate, 100 μM non-essential amino acids and 10% heat inactivated FCS. A549 cells were seeded onto fibronectin coated plates for all assays. VACV MVs were propagated in BSC40 or RPE cells and purified from cell lysates through a sucrose cushion prior to band purification on a sucrose gradient, as described previously [20]. All viruses used in this study were based on VACV strain Western Reserve. WR E-EGFP, WR L-EGFP, WR EGFP-A4L, WR mCherry-A4L and WR L4R-mCherry EGFP-F17R [WR VP8-mCherry EGFP-F17] have been previously described [8,92,93]. WR VACV vA2.5Li and vG4Li, vFS-A19i were the kind gift of B. Moss (National Institute of Allergy and Infectious Diseases, National Institutes of Health, Bethesda, Maryland, USA) [34,56,59]

### Generation of recombinant vaccinia viruses

Recombinant viruses were generated by homologous recombination as previously described [20]. Recombinant dual fluorescent viruses were generated using plasmids based on pJS4 for insertion of a second copy of the LB candidate protein with a C-terminal EGFP fusion protein into the Tk locus of WR mCherry-A4 [94]. For the G4L dual fluorescent virus, the EGFP fusion was N-terminal. ΔA45R and ΔO2L viruses were generated by replacing the 5' 232 nucleotides of A45 (as previously described [55]) or the entire O2L ORF. In both cases pBSIIKS plasmids containing EGFP flanked by 300 bp upstream and downstream of the genomic region to be deleted were generated. The introduction of EGFP into the region of interest was used for selection of deletion viruses. Viruses were subjected to four rounds of plaque purification to homogeneity, and deletion of A45 and O2 confirmed by sequencing of virus stocks.

### Virion fractionation

Band purified MVs were fractionated into sub-viral constituents by incubation with 1% NP-40 and 50 mM DTT in 10 mM Tris, pH 9.0, for 30 min at 37°C with gentle perturbation as previously described [95]. The insoluble fraction was sedimented for 1 hr at 21,130 x g, 4°C and the supernatant (Sup 1; Membrane fraction) retained. The insoluble fraction was washed (Wash 1) by resuspension in 10 mM Tris pH 9.0 and sedimented for 1 hr at 21,130 x g at 4°C (Pellet 1; LB-Core). Both fractions were retained for analysis. For LB removal, LB-Core fractions were treated with trypsin (31.25, 62.5, 125, 250, 500, 1000 ng/ml) in 10 mM Tri,s pH 9.0 for 15 min,

37°C prior to addition of 400 µg/ml Soybean Trypsin Inhibitor. The insoluble fraction was sedimented 1 hr at 21,130 x g at 4°C (Pellet 2; Core) and, along with the soluble supernatant fraction (Sup 2), retained for analysis. For large scale fractionations prior to MS or EM analysis, 125 ng/ml Trypsin was used.

## Electron Microscopy (EM)

LB-Core and Core samples were prepared as described (see Viral Fractionation) and prepared for EM by fixation in 1.5% glutaraldehyde / 2% EM-grade paraformaldehyde (TAAB) in 0.1M sodium cacodylate for 45 min at RT. Briefly, samples were treated with reduced osmium, tannic acid, dehydrated through an ethanol series and embedded in epon resin [96]. Sections were collected on formvar coated slot grids, stained with lead citrate and imaged using a transmission electron microscope (Tecnai T12, Thermo Fisher Scientific) equipped with a charge-coupled device camera (SIS Morada; Olympus).

## Immunoblotting

Immunoblot samples were boiled in LDS DTT sample buffer (Novex, Life Technologies) for 5 min at 98°C. Proteins were separated on 4–12% or 12% Bis-Tris gels (Invitrogen), transferred to nitrocellulose membranes, and analysed using rabbit anti-A10L (1:1,000), anti-F17R (1:1,000), anti-A17L (1:500) in combination with horseradish peroxidase (HRP)-coupled secondary antibodies (1:2,000) and Immoblion Forte Western HRP substrate (Merck) or with IR-Dye secondary antibodies (Licor) (1:10,000). Blots were visualised using photographic film and X-ray developer or LiCor Odyssey. Relative quantification of immunoblots were performed using the "Analyse Gels" tool in Fiji. Pixel intensities across equal sized rectangles encompassing each protein band on the blot were analysed using the "Plot Lanes" function. The intensity of each quantified protein band was plotted as the percentage of the value of each protein in the "Whole VACV" lanes.

## Mass Spectrometry (MS)

Lysis and tryptic digestions were performed under pressure-cycling conditions using Barocycler NEP2320-45k (PressureBioSciences, South Easton, MA) as described previously [61]. Briefly, LB-Core and Core fractions were resuspended in 8 M urea containing 100 mM ammonium bicarbonate pH 8.2, 10% 2,2,2 trifluoroethanol, one tablet of phosphatase inhibitors cocktail (PhosStop, Roche) per 10 ml buffer. Lysis was performed under pressure cycling conditions (10s at 45 kpsi, 10s at 0 Kpsi, 297 cycles). Samples were sonicated three times for 30s and then rested for 1 min on ice and spun at full-speed in a bench top centrifuge for 5 min. Supernatants were treated with 10 mM TCEP for 20 min at 35°C and then 40 mM iodoacetamide in the dark at RT for 30 min. Samples were diluted to contain 6 M urea and digested with LysC 1/50 w/w under pressure cycling conditions (25s at 25 Kpsi, 10s at 0 Kpsi, 75 cycles). Samples were further diluted to 1.5 M urea and digested with 1/30 w/w trypsin was performed pressure cycling conditions (25s at 20 Kpsi, 10s at 0 Kpsi, 198 cycles). Samples were mixed by inversion overnight at 37°C and digestion stopped by the addition of trifluoroacetic acid to pH 2.0. Resultant peptides were desalted on a reverse phase C18 column (Waters corp., Milford, MA) and eluted with 40% acetonitrile (ACN), 0.1% TFA. The solvents were evaporated using a centrifuge evaporator device. Peptides from LB-Core and Core samples were resuspended in 2% ACN, 0.1% formic acid subjected to direct LC-MS/MS analysis.

Peptide samples from LB-Core and Core samples were separated by reversed-phase chromatography on an ultra-high-pressure liquid chromatography (uHPLC) column (75 µm inner diameter, 15 cm, C18, 100 Å, 1.9 µm, Dr Maisch, packed in-house) and connected to a nano-

flow uHPLC combined with an autosampler (EASY-nLC 1000, Thermo Scientific). The uHPLC was coupled to a Q-Exactive Plus mass spectrometer (Thermo Scientific) equipped with a nanoelectrospray ion source (NanoFlex, Thermo Scientific). Peptides were loaded onto the column with buffer A (99.9% $H_2O$, 0.1% FA) and eluted at a constant flow rate of 300 nl per min over a 90 min linear gradient from 7% to 35% buffer B (99.9% ACN, 0.1% FA). After the gradient, the column was washed with 80% buffer B and re-equilibrated with buffer A. Mass spectra were acquired in a data-dependent manner, with an automatic switch between survey MS scan and MS/MS scans. Survey scans were acquired (70,000 resolution at 200 m/z, AGC target value $10^6$) to monitor peptide ions in the mass range of 350–1,500 m/z, followed by higher energy collisional dissociation MS/MS scans (17,500 resolution at 200 m/z, minimum signal threshold 420, AGC target value $5 \times 10^4$, isolation width 1.4 m/z) of the ten most intense precursor ions. To avoid multiple scans of dominant ions, the precursor ion masses of scanned ions were dynamically excluded from MS/MS analysis for 10 s. Singly charged ions and ions with unassigned charge states were excluded from MS/MS fragmentation.

COMET (release 2015.01 rev. 2) [97] was used to search fragment ion spectra for a match to tryptic peptides allowing up-to two missed cleavage sites from a protein database, which was composed of human proteins (SwissProt, v57.15), Vaccinia virus proteins (UniProt, strain Western Reserve, v57.15), various common contaminants, as well as sequence-reversed decoy proteins. The precursor ion mass tolerance was set to 20 ppm. Carbamidomethylation was set as a fixed modification on all cysteines. The PeptideProphet and the ProteinProphet tools of the Trans-Proteomic Pipeline (TPP v4.6.2) [98] were used for probabilistic scoring of peptide-spectrum matches and protein inference. Protein identifications were filtered to reach an estimated false-discovery rate of ≤1%. Peptide feature intensities were extracted using the Progenesis (v2.0) LC-MS software (Nonlinear Dynamics). Protein fold changes and their statistical significance between paired conditions were tested using at least two fully tryptic peptides per protein with the MSstats library (v1.0) [99]. Resulting p-values were corrected for multiple testing with Benjamini-Hochberg method [100]https://paperpile.com/c/s7nFaT/WYQlH.

### STRING analysis

The LB human protein-protein interaction network analysis was performed using the STRING database at high stringency (0.700), clusters were generated using kmeans clustering [8] (https://string-db.org).

### Flow cytometry

Cells were detached with 0.05% trypsin-EDTA, trypsin inactivated by addition of 5% FBS and fixed with 1% Formaldehyde in PBS. Samples were analysed for EGFP using a Guava easyCyte HT flow cytometer and InCyte software (Millipore).

### Core-LB preparation, immunofluorescence staining and SIM imaging

Band purified A4L-EGFP virions were subjected to fractionation as described under "virion fractionation". The Core-LB fraction was resuspended in 10mM Tris pH 9.0 and bound to coverslips for 30 min at room temperature. Bound Core-LBs were fixed, washed twice with PBS, and blocked in 5% BSA/PBS for 1h prior to antibody labelling. Virions were labelled with anti-Tomm20 (SCB sc-17764, mouse) or anti-RPL17 (RPL23; SCB sc-515904, mouse) (1:200) overnight at 4˚C. After washing with PBS, secondary anti-mouse Alexa Fluor 647 (ThermoFisher) was added for 1h at room temperature. Slides were imaged using Elyra, SIM using 5 phase shifts and 3 grid rotations with 488nm and 647nm lasers.

## Super-resolution imaging and correlative light and electron microscopy

Band purified MVs diluted in 20 μl 1 mM Tris pH 9.0 were sonicated and vortexed prior to being placed on coverslips for 30 min. Bound virus was fixed with 4% formaldehyde. When required, samples were permeabilized for 30 min with 1% TritonX-100 in PBS and blocked with 5% BSA (Sigma), 1% FCS for 30 min and immune-stained with anti-GFP nanobody (Chromotek) conjugated in-house to AlexaFluor647-NHS (Invitrogen) in 5% BSA, PBS at 4˚C overnight. Coverslips were washed three times with PBS prior to mounting. Both mounting and imaging by STORM and SIM and Virusmapper modelling have been described previously [74]. Correlative super-resolution light and electron microscopy imaging of MVs was performed as previously described [101].

## Viral yield assay

A549 cells were infected with WT VACV (MOI 1) for 1 h. Cells were treatment with 0–200 μM TBHP as indicated and incubated at 37˚C, 5% $CO_2$ for 18 h. Cells were collected and resuspended in 1 mM Tris pH 9.0. MVs were titred on BSC40s. Forty-eight h pi, cells were fixed, and viral plaques visualized using 0.1% crystal violet/ 2% formaldehyde. Titres were normalised to 0 μM TBHP condition.

## Cytotoxicity assay

A549 cells were treated with 0–200 μM TBHP for 8 h or 18 h. Cytotoxicity was measured using the Pierce LDH Cytotoxicity Assay Kit (ThermoScientific) according to the manufacturer's instructions. Absorbance was measured at 490 nm and 650 nm using a VERSA max microplate reader (Molecular Devices).

## Viral gene expression assays

For flow cytometry experiments, A549 cells were infected with WR E-EGFP (MOI 1) or with WR L-EGFP (MOI 3). At 1 h pi cells were treated with 0–200 μM TBHP for 8 h followed by flow cytometry analysis.

## Reverse transcription–quantitative PCR (RT-qPCR)

For RT qPCR experiments, A549 cells were infected with WT VACV (MOI 3) for 1h and cells treated with 0, 100, 200 μM TBHP for 8h followed by total cell RNA extraction. Total RNA was extracted using the RNeasy Plus Mini kit (Qiagen) according to manufacturer's instructions. Single stranded cDNA was reverse transcribed from 500 ng of total RNA with SuperScript II Reverse Transcriptase (Invitrogen) and oligo(dT) primers (Invitrogen). cDNA was diluted 1:5 in water and 2 μl used as template for qPCR using Mesa blue MasterMix Plus for SYBR (Eurogentec) and a CFXConnect Real-Time System (BioRad) PCR machine. The expression levels of the viral genes C11L (early), F17R (late), and the human gene GAPDH were measured using specific primers (Integrated DNA Technologies): C11L (5'-AAACACACACTGAGAAACAGCATAAA-3' and 5'-ACTATCGGCGAATGATCTGATTA-3'), F17R (5'-ATTCTCATTTTGCATCTGCTC-3' and 5'-AGCTACATTATCGCGATTAGC-3') GAPDH (5'-AAGGTCGGAGTCAACGGATTTG GT-3' and 5'-ACAAAGTGGTCGTTGAGGGCAATG-3'). Gene expression was normalised to GAPDH and then to the untreated condition for each gene.

## Fluorogenic detection of Oxidative stress

VACV MVs (MOI 5) were bound to A549 cells for 30 min at room temperature. Cells were shifted to 37˚C for 1 h. Cells were further incubated for 1 h or 8 h. Samples were treated with

250 nM CellROX green (Thermo-Fischer Scientific) or 5μM MitoSOX Red (Invitrogen), as indicated, for the final 30 min prior to analysis by flow cytometry. Where stated, cells were challenged with the indicated concentrations of TBHP for 1 h prior to flow cytometry analysis.

## Graphs and statistics

All graphs were generated in Prism 9 (Graphpad Software). Errors bars represent the standard error of the mean (SEM). Unpaired t-test and one-way anova analysis were performed in Prism 9. P values: $^{*}P< 0.05$, $^{**}P<0.005$, $^{***}P<0.0005$.

## Supporting information

**S1 Fig. VACV Fractionation quality control. (a)** Left: VACV MV fractionation and LB digestion conditions were optimised using WT VACV MVs. Virion membranes (Sup. 1) were removed by treatment with NP-40 + 50 mM DTT. To assess optimal conditions for LB removal, LB-Core samples (Pellet 1) were treated with various trypsin concentrations (31.25, 62.5, 125, 250, 500, 1000 ng/ml). The corresponding soluble (Sup. 2) and insoluble Core samples (Pellet 2) were retained. Viral fraction samples were analysed by immunoblotting against A17L (membrane protein), F17R (LB protein) and A10L (core protein). Right: the percentage of A10L and F17R residing in Pellet 2 (Core) upon increasing trypsin concentration was quantified using imageJ 'analyse gels' tool (see materials and methods). A single example image and quantification are displayed. A trypsin concentration of 125 ng/ml was selected, from multiple experiments, for preparing MS samples **(b)** Left: Representative immunoblot of A17L (membrane protein), F17R (LB protein) and A10L (core) from a large-scale fractionation performed, as determined in (a), for MS analysis. Right: Quantification of the percentage of A17L, F17R and A10L in Sup. 1 (Membrane), Pellet 1 (LB-Core) and Pellet 2 (Core) samples (n = 3; ± SEM). **(c)** WT purified VACV MVs (Whole VACV) were fractionated as in (a). Pellet 1 (LB-Core) and Pellet 2 (Core) samples were washed after the trypsin treatment, fixed, sectioned and imaged by TEM (Scale bars = 1 μm).
(TIFF)

**S2 Fig. Schematic of VACV Membrane, LB and Core interactions.** The VACV-MV protein-protein interaction network established by Mirzakhanyan and Gershon (65) was used to build a schematic of the interactions of the LB candidate proteins identified by this study. Proteins are colour-coded according to subviral location as indicated.
(TIF)

**S3 Fig. Cellular LB candidate proteins co-purify with virions.** Immunoblot analysis of cellular proteins identified as not enriched (actin, tubulin) or enriched (Hist1, RPL17, SOD1 and Tomm20) in LBs by comparative quantitative MS (Fig 2 and S3 Table). The VACV purification protocol was performed on uninfected and infected cell lysates and samples collected for analysis at multiple stages [Post cell lysis (cell lysate/nuclear pellet), pellet from 36% sucrose cushion, virus band from 25–40% sucrose gradient, and final virus pellet]. Experiments were performed in duplicate and representative blots shown.
(TIF)

**S4 Fig. Overviews of Correlative SIM, STORM and EM. (a)** Correlative SIM / STORM of WR mCherry-A4L A19L-EGFP virions. The two viral particles shown in Fig 3E are boxed. **(b)** Correlative STORM / EM of WR mCherry-A4L A19L-EGFP virions immunolabelled with anti-GFP nanobody. STORM images of the lateral body protein were registered with EM micrographs. The two viral particles shown in Fig 3F are boxed. Scale bars = 1 μm.
(TIFF)

**S5 Fig. TBHP treatment is not cytotoxic (a)** LDH release assay for cytotoxicity of TBHP titration on A549s cells under the conditions used in Fig 4C. **(b)** LDH release assay for cytotoxicity of TBHP titration on A549s cells under the conditions used in Fig 4D and 4E
(n = 3 ± SEM).
(TIF)

**S1 Table. Viral proteins enriched in LB-Core and LB fractions.** List of viral protein peptides identified by MS (Sheet 1), enriched in Core-LB and Core fractions (Sheet 2), enriched in Core-LB fraction (Sheet 3) or enriched in Core fraction (Sheet 4).
(XLSX)

**S2 Table. Description of viral proteins enriched in LBs.** List includes Gene/Alternate gene name, protein/alternate protein name Uniprot ID, reported function, expression profile and subviral localization information.
(XLSX)

**S3 Table. Cellular proteins enriched in LB-Core and LB fractions.** List of cellular protein peptides identified by MS (Sheet 1), enriched in Core-LB and Core fractions (Sheet 2), enriched in Core-LB fraction (Sheet 3) or enriched in Core fraction (Sheet 4).
(XLSX)

## Acknowledgments

We thank V. Edwards and H. Mok for initial work on the project, V. Gould for her technical support throughout the project, P. Pereira and R. Henriques for advice on super-resolution imaging. We thank B. Moss (NIAID, NIH) for viruses and all members of the Mercer lab for helpful discussions and comments on the manuscript.

## Author Contributions

**Conceptualization:** Susanna R. Bidgood, Jerzy Samolej, Bernd Wollscheid, Jason Mercer.

**Data curation:** Karel Novy, Bernd Wollscheid.

**Formal analysis:** Susanna R. Bidgood, Jerzy Samolej, Karel Novy, David Albrecht, Jemima J. Burden, Jason Mercer.

**Funding acquisition:** Susanna R. Bidgood, Bernd Wollscheid, Jason Mercer.

**Investigation:** Susanna R. Bidgood, Jerzy Samolej, Karel Novy, Abigail Collopy, David Albrecht, Melanie Krause, Jemima J. Burden.

**Methodology:** Susanna R. Bidgood, Jerzy Samolej, Karel Novy, Abigail Collopy, David Albrecht, Melanie Krause, Jemima J. Burden, Bernd Wollscheid, Jason Mercer.

**Project administration:** Bernd Wollscheid, Jason Mercer.

**Resources:** Karel Novy.

**Software:** Karel Novy.

**Supervision:** Susanna R. Bidgood, Bernd Wollscheid, Jason Mercer.

**Validation:** Susanna R. Bidgood, Jerzy Samolej.

**Visualization:** Susanna R. Bidgood, Jerzy Samolej, Abigail Collopy, David Albrecht, Jemima J. Burden, Jason Mercer.

**Writing – original draft:** Susanna R. Bidgood, Jerzy Samolej, Karel Novy, David Albrecht, Jemima J. Burden, Bernd Wollscheid, Jason Mercer.

**Writing – review & editing:** Susanna R. Bidgood, Jerzy Samolej, Bernd Wollscheid, Jason Mercer.

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
