## [Editor Report · Decision Letter 0]

29 Sep 2021

Dear Dr. Mercer,

Having reviewed your submission I feel that your manuscript would be, in principle, suitable for PLoS Pathogens as long as revisions address the reviewer comments satisfactorily. On that point, your revision plan seems appropriate but I would urge you to make sure that you adequately address reviewer concerns about potential contamination of virus preparations, as this is where the primary novelty of your report lies. On the other hand, I also appreciate that pursuing the functionality of 5 viral and other host candidates identified in your screen may prove challenging and beyond the scope of this report, and your plans for attempts to address these comments seem reasonable to me. However, any final decision will depend on how the reviewers view the final revisions.

As such, I encourage you to proceed with a revision as outlined in your submission.

Best wishes,

Derek

We cannot make any decision about publication until we have seen the revised manuscript and your response to the reviewers' comments. Your revised manuscript is also likely to be sent to reviewers for further evaluation.

Sincerely,

Derek Walsh, PhD

Guest Editor

PLOS Pathogens

Klaus Früh

Section Editor

PLOS Pathogens

Kasturi Haldar

Editor-in-Chief

PLOS Pathogens

orcid.org/0000-0001-5065-158X

Michael Malim

Editor-in-Chief

PLOS Pathogens

orcid.org/0000-0002-7699-2064
---

## [Decision Letter · Decision Letter 1]

23 Mar 2022

Dear Dr. Mercer,

Thank you very much for submitting your manuscript "Poxviruses package viral redox proteins in lateral bodies and modulate the host oxidative response" for consideration at PLOS Pathogens. We have now received comments back from the 4 original reviewers of your manuscript. As you will see from their comments, there was overall enthusiasm from most but some reviewers raised significant lingering concerns regarding the lack of functionality of the proteins in question and the purity of virus preparations analyzed. Having further discussed this further with the reviewers, two key experiments were suggested that would address these concerns.

1) The authors could address the question of redundancy by expressing the candidate proteins in uninfected cells and treating with TBHP.

2) The authors could address the question of host cell LB proteins by using antibody and super resolution microscopy as they did for viral proteins.

As stated in our prior correspondence, I agree that it may be difficult to prove functionality of these proteins if there is redundancy or multi-factor complexity in how they operate. The reviewers and I felt that the characterization of host proteins as components of the LB is of sufficient interest to PLoS Pathogens in itself, if the SIM/STORM imaging approaches suggested could be added to address this shared concern of two reviewers. Based on the reviews, we are likely to accept this manuscript for publication, providing that you modify the manuscript according to the review recommendations.

Best wishes,

Derek

Sincerely,

Derek Walsh, PhD

Guest Editor

PLOS Pathogens

Klaus Früh

Section Editor

PLOS Pathogens

Kasturi Haldar

Editor-in-Chief

PLOS Pathogens

orcid.org/0000-0001-5065-158X

Michael Malim

Editor-in-Chief

PLOS Pathogens

orcid.org/0000-0002-7699-2064

Dear Dr. Mercer,

We have now received comments back from the 4 original reviewers of your manuscript. As you will see from their comments, there was overall enthusiasm from most but some reviewers raised significant lingering concerns regarding the lack of functionality of the proteins in question and the purity of virus preparations analyzed. Having further discussed this further with the reviewers, two key experiments were suggested that would address these concerns.

1) The authors could address the question of redundancy by expressing the candidate proteins in uninfected cells and treating with TBHP.

2) The authors could address the question of host cell LB proteins by using antibody and super resolution microscopy as they did for viral proteins.

As stated in our prior correspondence, I agree that it may be difficult to prove functionality of these proteins if there is redundancy or multi-factor complexity in how they operate. The reviewers and I felt that the characterization of host proteins as components of the LB is of sufficient interest to PLoS Pathogens in itself, if the SIM/STORM imaging approaches suggested could be added to address this shared concern of two reviewers.

Best wishes,

Derek

Reviewer Comments (if any, and for reference):

Reviewer's Responses to Questions

**Part I - Summary**

Reviewer #1: This article is well presented and the experiments are generally of high quality. The associated mass spectrometry data should prove a useful resource for the poxvirus field. The contribution of VACV redox proteins to virus replication has not been definitively demonstrated, but this study is likely to stimulate additional studies into the role of redox regulation in virus replication and the host-antiviral response.

Reviewer #2: All comments and raised concerns have been adequately addressed.

Reviewer #3: The major strength of the paper is identifying a few additional viral proteins that localize in LBs. The weakness of the paper is that the other conclusions regarding host proteins localized in LBs and a role for LB proteins in combatting the host oxidative response are not convincing.

Reviewer #4: The manuscript is a very interesting piece of work which addresses the protein composition of the two lateral bodies (LBs) that are striking features of vaccinia virions. Virions are delimited by a membrane and contain a discrete core which has a proteinaceous wall and encloses the genome and a repertoire of transcriptional mediators. The lateral bodies flank the core, and there is an emerging sense that they contain bioactive, signaling molecules that set up a cytoplasmic milieu that supports vaccinia infection. Using a combination of virion fractionation, proteomics, and high-resolution microscopy, the authors document and discuss the contents of the lateral bodies.

In this revised manuscript, the authors have addressed many of the critiques raised after the first submission, and the work is significantly improved. They highlight two groups of proteins in particular: cellular proteins and virally encoded proteins that are known to be redox regulators and cellular proteins, many of which are normally associated with mitochondria.

The work is of high quality and the discussion is thorough. Undoubtedly, future studies will provide a more rigorous test of which of these "candidate lateral body" proteins are truly important for infection and are truly enriched in lateral bodies - but the current work is an important contribution to the field. To combat the possibility that cellular components may merely contaminate virions and LBs, they perform a "mock" virion purification from uninfected cells. Although imperfect because of the lack of the "mass" of sticky virion components, it is a good effort and a good addition. At some level, the concerns about impurities are the "nature of the beast". Localizing the cellular proteins to the LBs by high-resolution SIM/STORM would be a good addition, and is well within the expertise of these authors.

**Part II – Major Issues: Key Experiments Required for Acceptance**

Reviewer #1: I have problems with interpretation of Figure S3, which is new data included in this revised manuscript. This figure supports that SOD1 and TOMM20 are, to some extent, found in the purified virus fraction (final pellet). Hist1 is not detected; this could reflect the differential sensitivity afforded by immunoblots versus mass spectrometry and its absence therefore neither supports nor refutes its specific association with virus particles. However, RLP17 is found at much higher abundance in the purified virus fraction (Final Pellet) in the uninfected cells than in the infected cells. To my eye it RLP17 is not at all visible in the infected cell sample (in contradiction to the statement on line 246-7). The authors don’t seem to have critically assessed the data presented in Fig S3. This data shows that ribosomal proteins (presumably in the context of intact ribosomes) can indeed be purified when following the procedure used to isolate virus particle. This should be mentioned explicitly, as it suggests that identification of any ribosomal protein in the virion fraction should be interpreted with caution.

Reviewer #2: The mansucript has been improved and essential comments have been addressed by the authors.

Reviewer #3: 1. The first major problem with the study relates to the inadequate purification of mature virions. Unlike many other viruses, vaccinia virus is isolated from cell extracts rather than the cell culture supernatant, making purification from host materials more difficult.

The authors responded to my criticism of their purification in 3 ways - (1) citing literature, (2) experimentally, and (3) by terminology.

With regard to (1), they justify their purification as “as field standard” but only in one of the half dozen or so papers cited was the purification for mass spectrometry and that was 16-years-old. The degree of purification needed depends on the proposed use and is very different for electron tomography and super resolution microscopy compared to mass spectrometry. Moreover, as mass spectrometry has improved and the depth of analysis increased, higher purity is needed. In a benchmark study not cited in the paper (Ngo, Mirzakhanayan and Gershon 2016) used two methods of purification, sucrose gradient and tartrate gradient, as the former separates by non-equilibrium rate zonal velocity and the latter by equilibrium buoyant density isopycnic banding. For each method hundreds of host proteins were detected by mass spectrometry. However, different host proteins were associated depending on the purification. By comparing the host proteins associated with mature virions by the two methods, the vast majority of cellular proteins were deemed non-packaged contaminants. Only ~60 overlapped by the two methods, and even these were still not excluded as contaminants. I did not notice any of the candidate LB host proteins of Bidgood et al in the that 60. In a related method, Resch et al. 2007 used two successive sucrose density gradient centrifugations followed by an isopycnic centrifugation and reported that many more host proteins were present prior to the isopycnic gradient (although the data were not shown).

(2) One of the two methods used by the present investigators to establish purity is electron microscopy. Vaccinia virus is ~ 300 nm. While the EM might exclude intact mitochondria, which can be comparable in size, it does not exclude ribosomal subunits (note that one of the “candidate LB proteins is ribosomal), nor does it exclude small membrane fragments or sticky proteins. The second experiment was to show that some of the cell proteins detected by mass spectrometry were also detected by Western blotting. However, unlike the viral core protein, which was enriched between the 36% sucrose cushion and the sucrose gradient, the host proteins were severely diminished. While this experiment confirmed the mass spectrometry identification using antibodies, it does not provide additional data regarding their presence as contaminants or constituents.

(3) Simply inserting the word “candidate” before the host proteins is insufficient.

If the authors want to consider any of the host proteins as LB or even candidate LB, they need to further purify the mature virions by isopycnic gradient centrifugation. It is regrettable that the authors did not consider this during the planning stage of this study.

2. The second major problem with this study is the lack of any evidence that the LB proteins are modulators of the host oxidative antiviral response, and no additional evidence was provided in the revision. A critical first experiment would be to show that viral protein synthesis is not needed for the response, but this was not done. Secondly, the authors found that preventing expression of the candidate LB proteins did not affect the oxidative response and concluded that this is because of redundancy of LB proteins. The simpler explanation, not offered, is that none of those LB proteins are involved. In fact, no evidence was brought forward to support the conjecture that the LB redox or any other LB proteins affect the oxidative status of the cells although that is implicit in the title of the paper.

Reviewer #4: One good experiment would be to perform SIM/STORM microscopy to test/verify the localization of a few of the mitochondrial proteins.

**Part III – Minor Issues: Editorial and Data Presentation Modifications**

Reviewer #1: Line 109: Typo – “viral phosphatase H1L” (space and H missing in text)

Figure 1C – It would be helpful to show the clustering of replicates, confirming that the replicates of the same conceptual 'experiment' are more similar to each other than to replicates from the other experimental condition. To put it simply, show the dendrogram for the columns too. Same for Figure 2a

Line 381 – You mean Fig S5a

Line 387 – You mean Fig S5b

Line 453 – If the data you are referring to is that presented in Fig S3, it doesn’t actually show Hist1 to be differentially purified in infected vs uninfected cells. Fig 3a doesn’t show differential purification between uninfected and infected cells. Please clarify what exact data you are referring to and be sure to comprehensively interpret the data (as per the above major comment).

Line 458 – You don’t use the possessive apostrophe for possessions of “it” – it’s is an abbreviation of “it is”. Either way, “of the VACV replicative niche” would be better.

Reviewer #2: All minor issues raised previously have been addressed.

Reviewer #3: (No Response)

Reviewer #4: Nothing of concern.

PLOS authors have the option to publish the peer review history of their article (what does this mean?). If published, this will include your full peer review and any attached files.

Reviewer #1: No

Reviewer #2: No

Reviewer #3: No

Reviewer #4: No

Figure Files:

Data Requirements:

Reproducibility:

References:

---

## [Editor Report · Decision Letter 2]

24 May 2022

Dear Jason,

We are pleased to inform you that your manuscript 'Poxviruses package viral redox proteins in lateral bodies and modulate the host oxidative response' has been provisionally accepted for publication in PLOS Pathogens.

Best regards,

Derek Walsh, PhD

Guest Editor

PLOS Pathogens

Klaus Früh

Section Editor

PLOS Pathogens

Kasturi Haldar

Editor-in-Chief

PLOS Pathogens

orcid.org/0000-0001-5065-158X

Michael Malim

Editor-in-Chief

PLOS Pathogens

orcid.org/0000-0002-7699-2064

Dear Jason,

I am happy to say that we feel your responses to the remaining reviewer comments and addition of new SIM data satisfactorily addressed the key question of host protein localized to LB's, and the inability to perform functional studies is understandable. Your manuscript is now accepted in principle pending editorial checks.

One thing I noticed in looking at Figure 3; panel 3g contains duplicated TEM images that is clearly a mistake during figure assembly and needs to be corrected.

Best wishes,

Derek
---

## [Editor Report · Acceptance letter]

21 Jun 2022

Dear Dr Mercer,

We are delighted to inform you that your manuscript, "Poxviruses package viral redox proteins in lateral bodies and modulate the host oxidative response," has been formally accepted for publication in PLOS Pathogens.

Best regards,

Kasturi Haldar

Editor-in-Chief

PLOS Pathogens

orcid.org/0000-0001-5065-158X

Michael Malim

Editor-in-Chief

PLOS Pathogens

orcid.org/0000-0002-7699-2064